# Anticoagulation with Factor Xa Inhibitors Is Associated with Improved Overall Response and Progression-Free Survival in Patients with Metastatic Malignant Melanoma Receiving Immune Checkpoint Inhibitors—A Retrospective, Real-World Cohort Study

**DOI:** 10.3390/cancers13205103

**Published:** 2021-10-12

**Authors:** Maximilian Haist, Henner Stege, Saskia Pemler, Jaqueline Heinz, Maria Isabel Fleischer, Claudine Graf, Wolfram Ruf, Carmen Loquai, Stephan Grabbe

**Affiliations:** 1Department of Dermatology, University Medical Center of the Johannes-Gutenberg University Mainz, 55131 Mainz, Germany; Henner.Stege@unimedizin-mainz.de (H.S.); Saskia.Pemler@unimedizin-mainz.de (S.P.); Jaqueline.Heinz@unimedizin-mainz.de (J.H.); mariaisabel.fleischer@unimedizin-mainz.de (M.I.F.); Carmen.Loquai@unimedizin-mainz.de (C.L.); Stephan.Grabbe@unimedizin-mainz.de (S.G.); 2Center for Thrombosis and Hemostasis, University Medical Center of the Johannes-Gutenberg University Mainz, 55131 Mainz, Germany; grafc@uni-mainz.de (C.G.); ruf@uni-mainz.de (W.R.)

**Keywords:** immunotherapy, immune checkpoint inhibitors, anticoagulation, factor Xa inhibitors, advanced melanoma, anti-tumor immunity, thromboprophylaxis, thromboembolic events

## Abstract

**Simple Summary:**

The advent of immune checkpoint inhibitors (ICI) improved the prognosis for patients with advanced melanoma. However, many patients do not benefit from ICI therapy due to primary and acquired resistance. Observations in murine systems suggested that coagulation factor Xa impedes anti-tumor immunity and that the oral FXa-inhibitor (FXa-i) rivaroxaban might synergize with ICI. In this retrospective study, we could demonstrate that concomitant treatment with anticoagulants did not impact the objective response rate, progression-free survival, or overall survival of stage IV melanoma patients who were treated with ICI. Remarkably, however, patients receiving concomitant treatment with FXa-i during initial ICI therapy showed a significantly improved objective response rate and progression-free survival as compared to patients not receiving anticoagulation or patients treated with other anticoagulants, such as heparins or vitamin K antagonists. Hence, our data suggest that FXa-i may augment ICI therapy, while patients who received FXa-i were not more likely to encounter bleeding complications.

**Abstract:**

Immune checkpoint inhibitors (ICI) significantly improved the prognosis of advanced melanoma patients. However, many patients do not derive long-term benefit from ICI therapy due to primary and acquired resistance. In this regard, it has been shown that coagulation factors contribute to cancer immune evasion and might therefore promote resistance to ICI. In particular, recent observations in murine systems demonstrated that myeloid-derived factor Xa (FXa) impedes anti-tumor immunity in the tumor microenvironment and that the oral FXa inhibitor (FXa-i) rivaroxaban synergizes with ICI. The synergistic effect of FXa inhibitors with clinical ICI therapy is unknown. We performed a retrospective study of 280 metastatic melanoma patients who were treated with ICI and stratified them for concomitant anticoagulation (AC) by medical chart review. Data on baseline patient characteristics, specific AC treatment, ICI therapy, other tumor-targeting therapies, and clinical outcomes were analyzed. Of 280 patients who received ICI, 76 received concomitant AC during initial ICI therapy. Patients on AC were treated either with heparins (*n* = 29), vitamin K antagonists (VKA) (*n* = 20), or FXa-i (*n* = 27). Patients requiring AC during ICI therapy showed no significantly reduced objective response rate (ORR) (*p* = 0.27), or progression-free (PFS; median PFS 4 vs. 4 months; *p* = 0.71) or overall survival (OS; median OS: 39 vs. 51 months; *p* = 0.31). Furthermore, patients who underwent AC did not show significantly more bleeding complications (*p* = 0.605) than those who were not anticoagulated. Remarkably, stratification of patients by the class of AC revealed that patients receiving FXa-i were more likely to obtain CR (26.9 vs. 12.6%, *p* = 0.037), and showed better ORR (69.2 vs. 36.4%, *p* = 0.005), PFS (median PFS: 12 months vs. 3 months; *p* = 0.006), and OS (median OS not reached vs. 42 months; *p* = 0.09) compared to patients not receiving FXa-i. Patient demographics and tumor characteristics in this patient subcohort did not significantly differ from patients not on FXa-i. In summary, our study provides first clinical evidence that the clinical application of FXa-i may enhance the efficacy of ICI therapy via the restoration of anti-tumor immunity, while patients who received FXa-i were not more likely to encounter bleeding complications.

## 1. Introduction

Immunotherapy with immune checkpoint inhibitors (ICI) has emerged as a promising treatment for many different types of cancer [1] and demonstrated strong and durable tumor regressions even at advanced cancer stages [1]. Melanoma was the first tumor in which an ICI showed breakthrough success [2,3]. The efficacy of ICI has been documented both by an increase in the median survival and by the occurrence of long-term survivors [4,5,6]. Despite the observation of durable clinical responses for a subset of melanoma patients, the majority of cancer patients do not derive long-term benefit from ICI therapy, which has been attributed to an intrinsic or acquired resistance [7]. Hence, there is a vital need to delineate molecular characteristics of ICI resistance [8,9,10], and to identify new targets that augment the efficacy of ICI, optimize patient selection, and overcome mechanisms of ICI resistance.

In this context, immune evasion in the tumor microenvironment (TME) is considered a crucial barrier for effective immunotherapy. Recent studies suggest that coagulation factors contribute to tumor immune evasion and might promote resistance to ICI therapy [11,12]. The coagulation system is a major pathway of innate defense, limiting infections and supporting immunity during the restoration of tissue injuries. However, many aspects of the coagulation system can be exploited by tumor cells to promote tumor progression and metastasis [12]. In this regard, it has been shown that the upregulation of the coagulation-initiating tissue factor (TF) on cancer cells, the activation of platelets, and platelet-leukocyte interactions facilitate tumor cell survival and metastasis [12,13,14]. Due to the procoagulant properties of tumor cells, advanced malignancies are commonly associated with a higher risk of thrombotic diseases [15,16]. In addition to tumor cells increasing their survival by thrombin generation [17], the activation of protease-activated receptors (PAR), and specifically PAR2 via cancer cell-expressed TF and coagulation factor VIIa or coagulation factor Xa (FXa), directly contribute to tumor progression [18]. Consistent with these observations, recent preclinical data demonstrated that myeloid-cell-synthesized FXa impedes anti-tumor immunity in the TME via the activation of PAR2, which in turn directly promotes tumor progression independent of the blood clotting cascade [12,18]. Because rivaroxaban, a direct oral anticoagulant (DOAC) targeting FXa, enhanced the infiltration of cytotoxic T cells (CTL) and dendritic cells (DC) to the tumor site [12], it has been proposed that targeting this pathway by oral FXa inhibitors (FXa-i) provides synergistic effects with ICI therapy.

Aligning with the observations from Graf et al. indicating that anticoagulation (AC) with FXa-i might restore ICI efficacy [12], preclinical data from Metelli and coworkers [11] provided additional mechanisms for the clinical use of AC with thrombin inhibitors, which shape the TME towards favorable antitumor immunity and might also enhance ICI efficacy. In particular, the authors showed that thrombin inhibition with dabigatran restored anti-tumor immunity and increased efficacy of anti-programmed-cell-death-protein (PD)-1 therapy by blocking the release of transforming growth factor β (TGF-β) [11].

Despite the robust preclinical data, these findings have not been translated into the clinical setting. With few exceptions in tumor subtypes, AC, in general, does not appear to be associated with an improved outcome in tumor patients [19,20]. Investigating the role of AC in the outcome of patients with non-small-cell lung cancer (NSCLC) who have received anti-PD-1 or anti-PD-1-ligand (PD-L1) treatment, Nichetti et al. found no effect of AC on the progression-free survival (PFS) or overall survival (OS) [16]. AC at any time in combination with ICI also showed no benefit in a larger cohort of advanced cancer patients [19]. However, given the strong heterogeneity of patients in terms of the primary tumor localization, the question as to whether the use of AC in general and the application of FXa or thrombin inhibitors more specifically might impact the outcome of ICI treatment might be better addressed in defined tumor entities with unmet clinical needs in improvements of ICI therapy. The exigency to comprehensively analyze the clinical wirings between the use of (different) anticoagulants and ICI in patients with advanced melanoma is highlighted when taking into account the increased risk of thromboembolic events (TEE) in patients treated with ICI and the TEE-associated adverse survival [21].

In this study, we examined a real-world cohort of 280 stage IV melanoma patients who received ICI and stratified their treatment outcome to concomitant use of anticoagulants. We first compared the outcomes of patients who received concomitant AC during initial ICI therapy with patients who were not on AC, irrespective of the type of AC. Next, we compared the outcome of patients stratified by the different classes of AC, namely heparins, vitamin K antagonists (VKA), and FXa-DOAC. Last, we evaluated the risk of bleeding complications in patients who underwent AC or not during ICI therapy, since advanced cancer patients have an increased risk of bleeding complications [22,23].

## 2. Materials and Methods

### 2.1. Patient Population

We retrospectively examined 280 subsequent patients with stage IV malignant melanoma who were treated with one or more lines of ICI at the University Medical Center Mainz between 2011 and 2021. Immunotherapy comprised combined checkpoint inhibitor therapy (ipilimumab (IPI) + nivolumab (Nivo)) or monotherapy (Ipi, Nivo, pembrolizumab (Pembro)). Patients enrolled in blinded randomized controlled trials were excluded due to uncertainty of therapy assignment, as well as patients who received ICI in an adjuvant setting. Patients receiving ICI therapy in open-label or single-arm interventional trials were eligible for inclusion.

Data on baseline demographics, tumor specifics (i.e., BRAF-status, tumor thickness, ulceration status, AJCC stage at treatment initiation, and localization of metastases), laboratory results, treatment with anticoagulants, reasons for AC, primary and secondary clinical outcomes, as well as data on systemic pretreatments, and initial ICI therapy (i.e., ICI agents, treatment duration, tumor progression, treatment cessation due to adverse events, AEs), treatment regimens after discontinuation of initial ICI, and the status of the patient at the time of data lock (March 2021) were collected by electronic chart review. To allow for the identification of patients receiving concomitant AC while on ICI therapy, we searched the electronic medical record of these 280 patients for any of the following anticoagulants: heparin products (low molecular weight heparins, LMWH; unfractionated heparin, UFH; enoxaparin; fraxiparin), VKA, or FXa-i (apixaban, rivaroxaban, edoxaban). All patients who have received AC for at least 1 month in the course of initial ICI therapy were categorized as “on AC”, while those not receiving AC or receiving AC at a later time point (i.e., during subsequent antineoplastic therapy) have been categorized as “not on AC”. Furthermore, we excluded prophylactic, short-term AC from the analysis, which was defined as receiving LMWH for <14 days in the context of hospitalization. Clinical decisions regarding the prescription of AC were made independently of this study.

### 2.2. Primary Clinical Outcomes

We analyzed the impact of concomitant treatment with AC on the primary clinical outcomes of patients, which were defined as best overall response (BOR) to initial ICI therapy, objective response rate (ORR), disease control rate (DCR), PFS, and OS. BOR was defined as complete response (CR), partial response (PR), stable disease (SD), or progressive disease (PD) according to the revised Response Evaluation Criteria in Solid Tumors (RECIST) guidelines. ORR was defined as the proportion of patients with CR or PR, and DCR was the proportion of patients with CR, PR or SD.

### 2.3. Bleeding Complications

Secondary clinical outcomes included bleeding complications occurring in association with initial ICI therapy. Therefore, we identified all major and minor bleeding events that were documented in a time frame from the initiation of ICI until up to 6 months after ICI cessation, by electronic chart review. Bleeding events were categorized as major bleeding, which comprised fatal bleeding, symptomatic bleeding in a critical organ, and/or bleeding causing a decrease of hemoglobin level by 2 g/dL, or leading to a transfusion of two or more units of whole blood or red blood cells [24]; or minor clinically relevant bleeding [25]. Minor clinically relevant bleeding complications did not meet the criteria for a major bleed but prompted a clinical response, such as hospital admission, change of AC therapy, or medical intervention by a healthcare professional.

### 2.4. Statistical Analysis

Descriptive statistics were used to analyze the baseline characteristics of the study population. Treatment duration was calculated as the period between initial drug administration and treatment discontinuation. PFS was calculated from ICI start to the date of radiological or clinical disease progression, last follow-up, or death from any cause. OS was calculated from ICI start to the date of death or last follow-up. Chi-square test was used to assess the association between the AC status or the different classes of AC and BOR, ORR, and DCR. Confidence intervals (CI) of 95% for categorical variables were calculated using the Clopper–Pearson method. Testing for equality between patients on AC and patients not receiving AC was performed using student’s *t*-test or Chi-square test. Comparisons between continuous variables of the different AC classes were performed using ANOVA variance analysis. The association between bleeding complications and anticoagulant use was assessed using Fisher’s exact test.

We employed Kaplan–Meier survival plots to illustrate median OS and PFS probabilities and to explore the association between AC, PFS and OS. Survival curves were compared using log-rank tests. Median duration of follow-up was calculated using the reverse Kaplan-Meier method. Cox’s proportional hazards models were applied to identify the strongest predictors for survival analyses by adjusting for baseline characteristics, treatment regimen and AC status. Here, hazards ratios (HR) were provided with 95% confidence intervals (CI). Multivariate analysis was calculated for the significant (*p* ≤ 0.05) variables by the univariate test or by a priori selection for biological relevance to evaluate their conjoint, independent effects on PFS or OS. In all cases, two-tailed *p*-values were calculated and considered significant with values *p* < 0.05. SPSS (version 27, IBM, Ehningen, Germany), RStudio (Version 1.3.1093), and GraphPad PRISM (Version 5, San Diego, CA, USA) were used for all analyses.

## 3. Results

### 3.1. Patients’ Characteristics

A total of 280 patients (168 male and 112 female) who received at least one line of ICI therapy were included. The median age at initiation of ICI was 66 years. Overall, 116 patients (42.2%) were diagnosed with BRAF-mutant melanoma, while 114 patients received at least one line of systemic therapy before initial ICI administration. Systemic pretreatments comprised either chemotherapy, IFN-α-treatment, BRAF/MEK inhibitor (i) therapy or other (i.e., RNA vaccination). Median Breslow tumor thickness was 2.5 mm, and 51.1% of melanomas were ulcerated at primary diagnosis. Baseline LDH serum levels were elevated in 159 patients (66.8%) and 70 patients presented with LDH-levels elevated by 1.5-fold (29.4%).

Among 280 patients, 109 initially received combined ICI therapy, whereas 171 patients were given ICI monotherapy, which comprised nivolumab (*n* = 55), pembrolizumab (*n* = 93), or ipilimumab (*n* = 23). Notably, all patients showed distant metastasis at the time of initial ICI application. In 200 patients, ICI therapy was given in a first-line setting (71.4%). Median duration of initial ICI therapy was 4.0 months (95% CI: 1.0–24.9 months), with 67 patients ceasing ICI therapy due to severe AEs. Initial ICI treatment was ongoing at the time of database lock in 21 cases (7.5%). At the time of data analysis, 220 patients (78.5%) had shown tumor progression upon initial ICI treatment, with 91 patients developing melanoma brain metastases (MBM; 32.9%) and 92 patients presenting with new liver metastasis (32.9%) during follow-up. Tumor progression required the re-initiation of subsequent treatments in 146 patients (52.1%). These patients received a median of 1 subsequent treatment line (range: 0–4) after initial ICI therapy, which comprised BRAF/MEKi (*n* = 62), chemotherapy (*n* = 16), ICI therapy (*n* = 107), or other (i.e., study medications, *n* = 9). During the overall observation period, 114 patients died (40.6%). Median follow-up time of the study cohort was 28.0 months (95% CI: 23.4–32.6).

Overall, 76 patients underwent continuous anticoagulation during initial ICI therapy (27.0%), and 42 received platelet aggregation inhibition with aspirin (15.0%). Patients receiving AC were treated either with heparins (*n* = 29), VKA (Marcumar; *n* = 20), or oral FXa inhibitors (*n* = 27). The most common indications for AC were deep vein thrombosis (*n* = 25), atrial fibrillation (*n* = 26), or pulmonary embolisms (*n* = 11). Further details on baseline patient characteristics and the subgroups stratified by the application of AC during ICI therapy are provided in Table 1.

No significant differences in terms of clinical and biological characteristics were observed between patients receiving AC or not, except for the age at initiation of ICI, PTT serum levels, and the therapeutic agents applied during initial ICI therapy. Specifically, it has been found that patients who received concomitant AC during initial ICI therapy were significantly older (70 vs. 63 yrs, *p* < 0.001) and less often received combined ICI therapy as compared to patients not receiving AC (26.3 vs. 43.6%, *p* = 0.02).

### 3.2. Factors Associated with Disease Progression and Survival upon ICI Therapy

To allow for a comparison of the investigated cohort with melanoma cohorts described in previous studies, we used Cox regression analysis to identify clinical and biological factors affecting disease progression upon ICI therapy. The best predictors of disease progression in univariate analysis were elevated serum LDH levels, BRAF mutation, and the presence of MBM, while the application of ICI in a first-line setting, a longer treatment duration, objective response to ICI, and the absence of bleeding complications during ICI therapy were associated with a longer PFS (Appendix A). Cox regression analysis also revealed an association of elevated serum LDH levels and MBM with OS, whereas the response to initial ICI, treatment duration > 4 months, and the absence of bleeding events during ICI were associated with a better OS (Appendix A).

Given the number of target events and the biological rationale, we included BRAF-status, LDH-serum levels, the presence of MBM, and other covariates statistically associated either with PFS or OS in a multivariate analysis (Appendix A). This multivariate model confirmed an independent association of serum LDH levels (HR: 1.39, 95% CI: 1.03–1.88, *p* = 0.081), MBM (HR: 0.615, 95% CI: 0.45–0.85, *p* = 0.001), treatment duration (HR: 0.23, 95% CI: 0.16–0.33, *p* = 0.003), and objective response to initial ICI (HR: 0.19, 95% CI: 0.11–0.33. *p* < 0.001) with PFS. Of note, BRAF wildtype and ICI monotherapy only showed a non-significant trend for PFS. The multivariate model also confirmed a significant association of LDH serum levels, the presence of MBM, BOR to initial ICI therapy, and the event of bleeding complications during ICI with OS (Appendix A). Survival curves and results from log-rank tests for the variables affecting PFS and OS in this study cohort are shown in Appendix A.

### 3.3. Impact of Concomitant Anticoagulation Treatment on Survival in Melanoma Patients Receiving ICI

Next, we investigated the impact of concomitant AC on the response to initial ICI therapy. Our analysis unveiled no significant differences in the BOR of patients who did or did not receive AC (*p* = 0.576). Moreover, we found no significant differences in the ORR (*p* = 0.270) or DCR (*p* = 0.946) between patients receiving AC and those who did not (Table 2). In particular, patients receiving AC had an ORR of 43.8% (95% CI: 32.2–55.9%), whereas those not on AC had an ORR of 38.4% (95% CI: 32.0–45.9%).

Aligning with the results from univariate Cox regression analysis (Appendix A), log-rank tests showed no significant association between PFS (median PFS: 4 vs. 4 months, *p* = 0.72, HR: 0.95, 95% CI: 0.71–1.28) or OS (median OS: 39 vs. 51 months, *p* = 0.31, HR: 0.81, 95% CI: 0.54–1.22) and concomitant anticoagulation (survival curves are shown in Figure 1). In a multivariate analysis, which has been performed after adjusting for potential confounders such as the higher age found in patients on AC, we also observed no association neither between PFS (HR: 0.86, 95% CI: 0.59–1.26, *p* = 0.44) or OS (HR: 0.78, 95% CI: 0.47–1.3, *p* = 0.34) and AC status (Appendix A). Due to the observation that patients receiving concomitant AC were a heterogeneous group in terms of clinical outcomes, we further distinguished patients into three categories stratified by the specific class of anticoagulant received. Here, regression analysis revealed that patients treated with FXa DOAC showed significantly better outcomes in terms of PFS compared to patients receiving other anticoagulants (i.e., heparins and VKA) (HR: 0.36, 95% CI: 0.20–0.64, *p* < 0.001), but also compared to all other patients not receiving FXa DOACs (HR: 0.44, 95% CI: 0.25–0.78, *p* = 0.0056).

### 3.4. Concomitant Treatment with Factor-Xa Inhibitors Is Associated with a Better Response and Longer PFS upon Initial ICI Therapy

We then analyzed the treatment outcomes stratified by the class of AC received by patients while on initial ICI therapy. Using the Chi-square test, we observed that patients receiving FXa DOACs showed a significantly higher ORR (69.2%, *p* = 0.002) and DCR (84.6%, *p* = 0.004) as compared to patients receiving other anticoagulants (Table 3 and Figure 2). Notably, patients receiving FXa DOACs were more likely to obtain a CR (*p* = 0.035), and also showed a better ORR (*p* = 0.004) and DCR (*p* = 0.028) as compared to patients not receiving concomitant AC.

Given the preliminary associations of the AC categories with a better ORR, we further explored the impact of FXa DOACs on survival. With regard to the PFS, Kaplan–Meier analysis revealed that patients receiving FXa DOACs had a significantly longer PFS (median PFS: 12 months; 95% CI: 3.5–20.5 months, *p* < 0.001) compared to patients receiving other anticoagulants, such as heparins or VKA (median PFS: 2 months, 95% CI: 0.75–3.25 months). Patients on FXa DOACs also showed a longer PFS as compared to patients not on AC (median PFS: 4 months, 95% CI: 2.9–5.1 months, *p* = 0.016) (Figure 3A), whereas patients receiving heparins showed the shortest PFS (median PFS: 2 months) of all patient subcohorts.

Results from our survival analysis further showed that patients receiving concomitant FXa DOAC treatment presented with a longer OS (median OS: not reached, *p* = 0.007) as compared to patients given other AC, that is, heparins (median OS: 17 months, 95% CI: 10.6–23.4 months) and VKA (median OS: 39 months, 95% CI: 16.8–61.2 months). Despite an observed favorable trend, patients with concomitant FXa DOAC therapy did not present with a better OS as compared to patients not receiving AC (median OS: 51 months, 95% CI: 30.6–71.3 months, *p* = 0.152) (Figure 3B). This finding is consistent with Cox regression analysis, which demonstrated that patients receiving FXa DOACs had a smaller risk of death (HR: 0.49; 95% CI: 0.22–1.12); however, this association was below statistical significance both in univariate (*p* = 0.09) and multivariate analysis (*p* = 0.55) (Appendix A). On the other hand, patients receiving heparins during initial ICI therapy showed a significantly shorter median OS as compared to patients not receiving AC (*p* = 0.003).

### 3.5. Patient Characteristics Receiving Concomitant FXa-Inhibitor Treatment

Patients receiving concomitant treatment with FXa DOACs were predominantly male (59.3%), of older age (median: 69 years), and BRAF-negative (63.0%), with 70.3% of patients showing elevated serum LDH levels at treatment initiation, and mainly received either combined ICI therapy (32.1%) or anti-PD-1 monotherapy (64.3%). Notably, patients on FXa DOACs therapy underwent systemic pretreatments in 32.1% of cases, developed MBM in 38.5%, and received ICI in the first-line setting in 85.2% of cases. Furthermore, our data revealed that there was an increased use of FXa DOACs over time (median: July 2018), despite no statistically significant differences in terms of the time of AC treatment initiation have been observed as compared to the other patient subcohorts (heparins: March 2017; VKA: February 2018; no anticoagulation: December 2017; *p* = 0.123). We next analyzed whether patient demographics, and tumor and treatment specifics of patients receiving concomitant FXa DOAC therapy differed from the other patient subcohorts. Our data revealed that patients on concomitant FXa DOACs did not significantly differ from other patients in terms of tumor and treatment specifics, except for being of older age (median age: 69 yrs vs. 63 yrs; *p* = 0.009) and having received initial ICI for a longer period; however, this association was below statistical significance (*p* = 0.063, Appendix A). Moreover, we could observe that patients on FXa DOAC therapy less often required treatment discontinuation until the time of data-lock (*p* = 0.013). In particular, patients were less likely to be withdrawn from initial ICI therapy due to PD (*p* = 0.016), but cessation due to AEs has not been reduced (Appendix A). Further, patients receiving FXa DOAC did not experience more immune-related AEs as compared to patients not on AC (33.3% vs. 30.4%). Last, patients with concomitant FXa-i received fewer lines of subsequent therapies as compared to patients prescribed with other anticoagulants or patients not receiving AC (Appendix A).

### 3.6. Treatment with Anticoagulants Does Not Increase the Event of Bleeding Complications in Melanoma Patients Treated with ICI

Major adverse effects of prophylactic and therapeutic AC are bleeding complications, which were identified to be significantly associated with PFS and OS of metastatic melanoma patients, as indicated by Cox regression analysis (Appendix A). Therefore, we compared the risk of bleeding complications between patients receiving concomitant AC and those not on AC. Results showed that patients receiving AC during ICI therapy were not more likely to suffer a major or clinically relevant minor bleeding complication at any point during initial ICI therapy as compared to patients not on AC (Table 4; 9.2% vs. 6.9%; *p* = 0.61). Overall, 21 bleeding complications were documented during the observation period of initial ICI therapy. Thereof, 7 patients (2.5%) experienced major bleeding (3 among patients with concomitant AC (4.0%) and 4 among patients not receiving AC (1.9%)), whereas 14 (3.9%) patients showed minor bleeding complications while receiving ICI (5.3% for patients with concomitant AC vs. 4.9% for patients not receiving AC). Notably, there was no significant difference in the incidence of major or clinically relevant minor bleeding complications between the different categories of AC (*p* = 0.661).

## 4. Discussion

The advent of ICI led to profound tumor responses in some patients with advanced melanoma [28]. However, only a subset of patients eventually showed a sustained clinical response upon receiving ICI [7]. Recent evidence suggests that coagulation factors contribute to tumor immune evasion and resistance to immunotherapy [11,12]. In particular, we and others could demonstrate in preclinical studies that FXa and thrombin contribute to immune evasion [11,12]. This is in accordance with the well-established link between the coagulation system and cancer progression [13,15]. More specifically, recent clinical observations indicate that the occurrence of TEE during ICI therapy might worsen the survival of melanoma patients. Therefore, it has been proposed that concomitant treatment with (specific types of) AC might synergize with ICI and thereby augment anti-tumor activity.

Here, we present the outcomes of 280 stage IV melanoma patients who were treated with ICI and analyzed the impact of concomitant anticoagulation therapy. To our knowledge, this is the first study to examine this issue specifically in a real-world cohort of patients with advanced melanoma. Here, we obtained several key findings that corroborate previous concepts of the pathophysiological role of the coagulation system in cancer progression.

First, our results revealed that concomitant AC (regardless of the type of AC treatment) did not improve response or survival (median PFS: 4 vs. 4 months, *p* = 0.72; median OS: 26 vs. 47 months, *p* = 0.31) for melanoma patients who simultaneously received initial ICI therapy. This finding was unexpected because randomized trials showed that the application of LMWH and DOACs reduced the risk of VTE in cancer patients and may thus also reduce the risk of VTE-associated mortality [29,30,31]. However, our data are consistent with previous studies and randomized phase III trials [32], which found no association of concomitant AC with prolonged survival in patients with NSCLC [16] or other advanced solid cancer entities [19,21]. Although the cancer-specific risk of VTE in the pre-ICI era was found to be smaller in melanoma patients [33], melanoma patients treated with ICI show similar rates of TEE- and VTE-associated mortality as compared to patients with other advanced cancers [27,34,35], providing a biological rationale for AC therapy [21].

In the absence of a biological rationale for our observation, we reasoned that patients receiving AC treatment might show worse prognostic properties in comparison to patients without AC. Aligning with previous studies our data unveiled that patients receiving AC were significantly older and less often received combined ICI therapy [16,19]. Moreover, patients receiving AC showed a trend towards a higher tumor burden as reflected by higher LDH serum levels upon ICI initiation. After adjusting for these potential confounders in multivariate analysis, our data yet revealed no significant differences in ICI therapy outcomes (PFS: HR: 1.12, *p* = 0.62; OS: HR: 0.91, 95%, *p* = 0.78) between patients receiving AC and those who did not, indicating that concomitant AC, in general, does not impact the efficacy of ICI in metastatic melanoma patients.

Second and most importantly, we demonstrated that the individual class of AC given during initial ICI therapy was significantly associated with treatment outcomes of metastatic melanoma patients. Owing to the strong heterogeneity in terms of treatment outcomes among patients receiving AC, we stratified the patients by the class of AC prescribed. Intriguingly, patients receiving FXa DOACs had a significantly better ORR (*p* = 0.005) and a longer PFS upon initial ICI therapy (median PFS: 12 vs. 3 months; *p* = 0.006) and showed a non-significant trend towards a better OS as compared to patients not receiving FXa-i (median OS: not reached vs. 42 months, *p* = 0.09). We find it remarkable that a statistically significant correlation between FXa inhibition and improved outcome of ICI therapy was apparent even with this relatively small patient cohort, which may suggest that the synergy between FXa DOACs and ICI is rather potent.

Notably, the beneficial effects of FXa DOACs with ICI were not seen in patients treated with VKA, which prevents the synthesis of functional prothrombin and FX in the liver and FX in macrophages [12]. This clinical evidence may suggest that additional Vitamin-K-dependent coagulation factors participate in the regulation of anti-tumor immunity.

Our finding is entirely consistent with results from preclinical studies, which demonstrated that specifically FXa DOACs augment anti-tumor immunity by promoting an accumulation of CTL and DC in the TME, while preventing the recruitment of regulatory T cells (Treg) and macrophage polarization towards an immunosuppressive phenotype [12]. Tumor-associated macrophages have been found to regulate angiogenesis in various mouse primary tumor models and correlate with microvessel density in human tumors [36]. The synergistic effect of FXa DOACs result from inhibiting the signaling function of macrophage-derived FXa that promotes immune evasion through PAR2. Only FXa DOACs with tissue penetrance reach FXa in this location and specifically target this cell-autonomous signaling pathway, whereas heparins, which depend on antithrombin for FXa inhibition, are largely restricted to the intravascular and perivascular space and were not shown to enhance ICI efficacy [12,37]. In accordance, patients receiving heparins apparently showed worse outcomes as compared to patients not receiving AC at all (median PFS 2 vs. 4 months, *p* = 0.003). Because these patients were more often prescribed heparins due to severe TEE, such as pulmonary embolisms, and had received more systemic pretreatments prior to initial ICI, our data do, however, indicate that these patients were in worse overall clinical condition in general. Due to the increased incidence of severe and/or fatal TEE in this patient subcohort, we reasoned that the occurrence of severe TEE might be an important confounder, particularly when interpretating the results of PFS analysis, since the occurrence of fatal TEE might potentially impact PFS irrespective of the biological response to ICI therapy. After adjusting for these potential confounders in multivariate analysis, we showed that the association between heparin treatment and the risk of disease progression was yet not significant (HR: 1.12, 95% CI: 0.72–1.73, *p* = 0.62) when compared to non-anticoagulated patients. This is consistent with recent data, which found no significant effect of heparin treatment on survival for advanced cancer patients treated with ICI [16,19].

Notably, patients given FXa DOACs did not significantly differ from patients receiving other classes of AC or patients not on AC, except for being of older age and showing a non-significant trend of having received ICI more often in a first-line setting. Therefore, it seems conceivable that the survival benefit found in this subcohort might even be underestimated, because patients with AC treatment in general, and FXa-i therapy in particular, tend to represent a cohort with negative prognostic properties as compared to patients not receiving AC, due to a higher rate of TEE and other comorbidities (i.e., atrial fibrillation, previous cardiovascular events, or VTE). However, patients receiving FXa-DOACs might yet show a favorable risk profile in terms of cardiovascular preconditions as compared to patients who have received heparins, since heparins were mainly used following acute, severe TEE, such as pulmonary embolism, whereas FXa DOACs were mainly used for atrial fibrillation or minor DVT events (see Appendix A). Furthermore, we have observed a yet non-significant trend towards an increased use of FXa DOACs over time, which might be considered an additional confounder, taking into account the introduction of more efficient treatment regimens over time.

Last, our results revealed that patients receiving concomitant AC did not show more bleeding complications (*p* = 0.61) or any immune-related AEs. These observations complement a recent retrospective study which observed no increased risk of intracranial hemorrhage during AC for patients with MBM [23], suggesting a potential safety of AC in melanoma patients receiving AC due to VTE. However, as for the limited number of bleeding events in our study, it has to be suspected that patients with bleeding events might have been primarily treated and documented elsewhere, which complicates the systematic documentation of bleeding events in our retrospective study. Therefore, the interpretation of our data on bleeding complications requires caution and needs further systematic investigation in prospective clinical studies.

Further limitations of our study are the retrospective, monocentric nature of the investigation, which adds an inherent selection bias within the cohort. Moreover, the heterogeneity in systemic pretreatments, the different classes of ICI initially administered, and subsequent treatment lines, as well as some missing data, such as tumor thickness or ulceration status for several cases, might have affected our results. We also acknowledge that patients in our retrospective analysis had worse outcomes as compared to previous ICI approval studies, which might be attributed to the frequent application of pretreatments and the fact that, in particular during the first years of our observation period, the administration of IPI was still a common scheme in the treatment of metastatic melanoma. Other limitations are the different schedules of AC treatment applied for melanoma patients (e.g., different durations of concomitant AC and changes in therapeutic and prophylactic dosages of AC) and the restriction of our analysis to initial ICI therapy.

In summary, we provide first clinical evidence for a synergy of FXa DOACs given during initial ICI therapy in patients with metastatic melanoma, which complements recent preclinical studies that demonstrated a synergy between ICI and FXa DOACs, but not heparin. This effect was restricted to FXa DOACs and not seen for other anticoagulants, suggesting that the inhibition of the prothrombotic effects of coagulation may not account for the observed effects of FXa DOACs. Although our study was exploratory in its nature and therefore does not allow for definitive conclusions, our data suggest a robust synergy between FXa DOACs and ICI and may therefore serve as a stimulus for further investigations in prospective studies. Based on our observations, such clinical studies investigating the efficacy of concomitant FXa DOAC treatment might be particularly relevant for patients who are at a high risk of VTE prior to the initiation of ICI-based immunotherapy, while these studies should further take into consideration the potential risk of bleeding complications. If these findings are confirmed, our observations may represent a valid option to prevent thrombotic events during ICI therapy and augment ICI efficacy.

## 5. Conclusions

This retrospective study including 280 stage IV melanoma patients provides first clinical evidence that the application of FXa-i may enhance the efficacy of ICI therapy via the restoration of anti-tumor immunity. In particular, we could show that concomitant treatment with FXa-i improved the objective response rate and progression-free survival upon first-line ICI therapy, while patients who received FXa-i were not more likely to encounter bleeding complications.

## Figures and Tables

**Figure 1 cancers-13-05103-f001:**
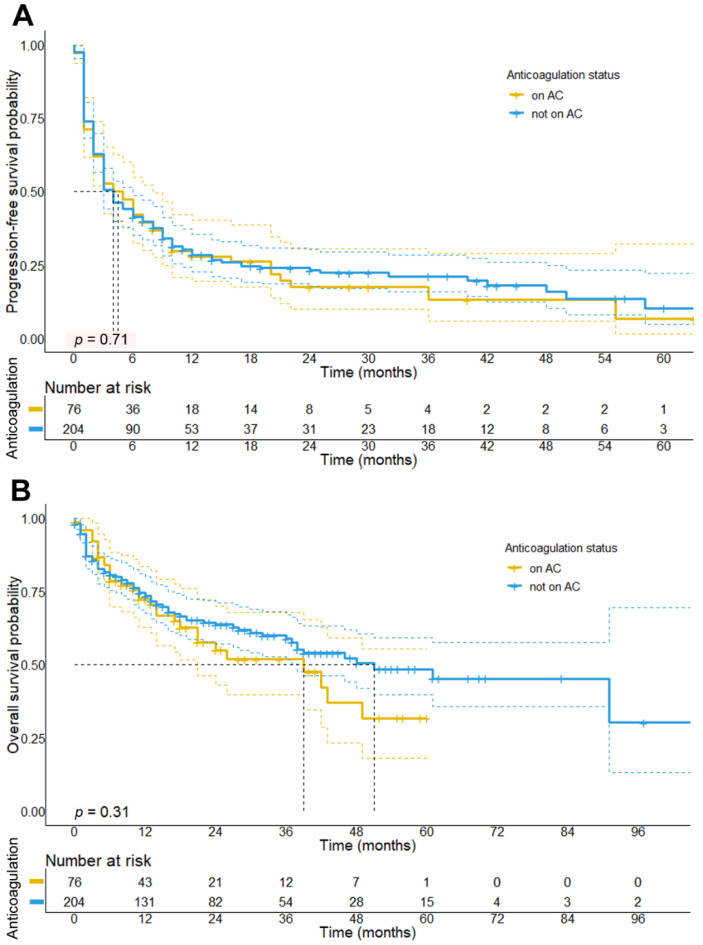
**Progression-free and overall survival in patients stratified by the concomitant treatment with anticoagulants.** Results show that patients receiving AC during initial ICI therapy did not have a significantly shorter PFS compared to patients not on AC (median PFS 4.0; 95% CI: 1.7–6.3 months vs. 4.0 months; 95% CI: 2.9–5.1 months; *p* = 0.705) (**A**). Neither did patients on AC show a significantly worse OS as compared to patients not on AC (median OS: 39.0; 95% CI: 18.4–59.6 months vs. 51.0 months; 95% CI: 30.6–71.3 months; *p* = 0.31), although there was a trend towards a shorter OS in this cohort of patients (**B**). The bold is used for the title of the figures.

**Figure 2 cancers-13-05103-f002:**
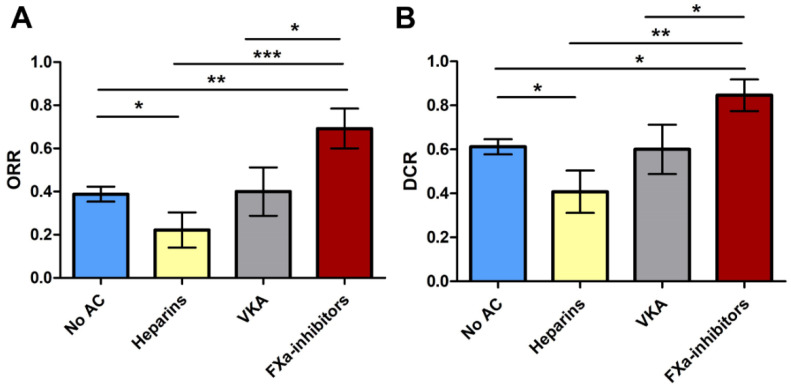
Bar chart showing the objective response rate (**A**) and disease control rate upon initial ICI therapy (**B**) stratified by the category of concomitant AC treatment. Patients receiving concomitant treatment with FXa inhibitors showed a significantly higher ORR and DCR as compared to other AC agents, such as heparins or vitamin K antagonists (VKA), as well as patients not receiving AC. Notably, patients on heparin treatment presented with the worst ORR and DCR as compared to the other categories. Abbreviations: ** p* < 0.05, *** p* < 0.005, **** p* < 0.001.

**Figure 3 cancers-13-05103-f003:**
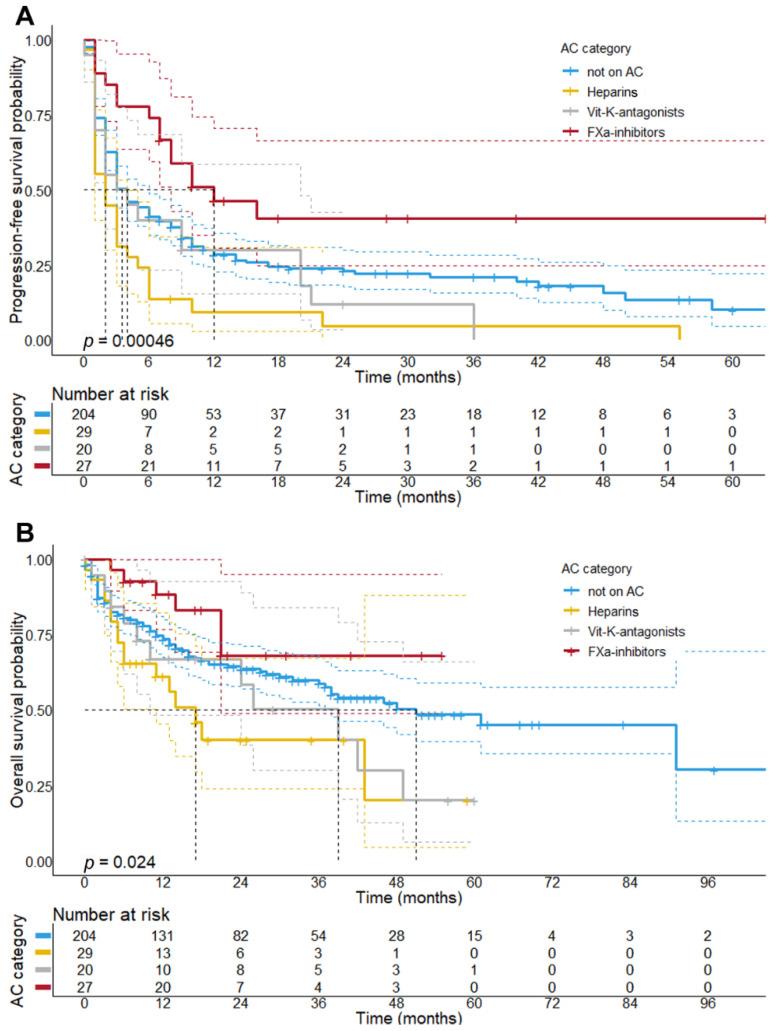
**Progression-free and overall survival of patients stratified by the category of concomitant anticoagulation.** Results revealed that PFS (*p* = 0.00046) and OS (*p* = 0.024) of patients undergoing initial ICI therapy significantly differed depending on the class of concomitant AC (*p*-values indicate statistically significant differences between the groups tested in this model). Particularly, patients receiving FXa-inhibitors (FXa-i) presented with the best PFS (median: 12 months) as compared to patients given heparins or VKA (median PFS: 2 and 3 months) during initial ICI therapy (*p* < 0.001). Further, patients not on AC did show a longer PFS compared to patients treated with heparins (*p* = 0.003), but a worse PFS as compared to patients receiving FXa-i (*p* = 0.016) (**A**). Log-rank test comparing the OS of patients stratified by AC category revealed a significantly shorter OS of patients receiving heparin treatment (median OS: 17.0 months, 95% CI: 10.6–23.4 months, *p* = 0.03), whereas concomitant treatment with VKA (median OS: 39 months; 95% CI: 16.8–61.2 months, *p* = 0.19) or FXa-i (median OS: not reached, *p* = 0.15) did not differ from patients not on AC (median OS: 51 months; 95% CI: 30.6–71.3 months) (**B**).The bold has been used for the image caption title.

**Table 1 cancers-13-05103-t001:** Baseline patient characteristics and treatment outcomes.

Clinicopathological Features	Not on Anticoagulation	On Anticoagulation	*p*-Value
Total number of patients	204	76	
Median age at initiation of ICI	63.0	70.0	**0.0006**
Gender			0.784
Female	121 (59.3%)	47 (61.8%)	
Male	83 (40.7%)	29 (38.2%)	
**Primary Tumor and Metastases**			
Median Breslow thickness (95% CI) ^1^	2.5 (0.7–12.7 mm)	2.4 (0.7–6.8 mm)	0.185
Ulceration ^2^	67/132 (50.8%)	28/54 (51.8%)	0.511
BRAF status ^3^	88/201 (43.8%)	28/74 (37.8%)	0.411
Elevated serum LDH levels (>245 U/L) ^4^	115/173 (66.5%)	44/65 (67.7%)	0.269
PTT at initiation of ICI ^5^	28.45 s	30.89 s	**<0.0001**
Systemic pretreatments prior to ICI - BRAF/MEKi therapy - IFNα therapy - Chemotherapy - Other	86 (42.1%) 48 37 11 5	28 (36.8%) 17 12 3 3	0.494
Melanoma brain metastases	65 (32.3%)	26 (34.6%)	0.774
Liver metastases	67 (32.8%)	25 (33.3%)	0.523
**Treatments**			
Initial ICI therapy			**0.020**
▪ Ipilimumab + nivolumab ▪ Nivolumab ▪ Pembrolizumab ▪ Ipilimumab	89 (43.6%) 39 (19.1%) 58 (28.4%) 18 (8.8%)	20 (26.3%) 16 (21.1%) 35 (46.1%) 5 (6.6%)	
Treatment line			0.883
- First - Not first	145 (70.4%) 59 (42.6%)	55 (72.4%) 21 (36.9%)	
Median treatment duration (range) ▪ Ongoing treatment ○ Number of patients ▪ Cessation due to toxicity ○ Number of patients ▪ Cessation due to PD ○ Number of patients	3.0 months (0–41) 14 months (7–41) 13 (6.4%) 3 months (1–23) 51 (25.1%) 2 months (0–8) 80 (39.2%)	4.0 months (0–38) 10 months (7–28) 8 (10.5%) 2 months (1–21) 16 (21.1%) 3 months (0–10) 28 (36.8%)	0.261
BOR to initial ICI therapy ^6^			0.569
- CR - PR - SD - PD - Could not be evaluated	30 (14.7%) 48 (23.5%) 45 (22.1%) 80 (39.2%) 1	9 (11.8%) 23 (30.3%) 13 (17.1%) 28 (36.8%) 3	
TP upon initial ICI	159 (77.9%)	61 (80.2%)	0.674
Median progression-free survival (95% CI)	4.0 months (2.9–5.1)	4.0 months (1.7–6.2)	0.705
VTE during initial ICI therapy ^7^	9 (4.4%)	26 (36.8%)	**<0.001**
ATE during initial ICI therapy ^8^	6 (2.9%)	6 (7.9%)	0.094
**Subsequent treatments**			
Patients receiving subsequent therapy Number of post-treatments ▪ Re-induction of ICI ^9^ ○ ORR ○ TP ▪ (Re)-induction of BRAF ± MEKi ^10^ ○ ORR ○ TP	103 168 78 33 (42.3%) 66 (84.6%) 45 18 (40.0%) 39 (86.6%)	43 68 29 11 (38.0%) 22 (75.9%) 19 6 (31.5%) 17 (89.5%)	0.900 0.673 0.587 0.764 0.611 0.614
**Follow-up**			
Median follow-up upon ICI initiation (95% CI)	30 months (24.4–35.6)	22 months (13.6–30.4)	0.055
Median overall survival (95% CI)	51 months (30.6–71.4)	39 months (18.4–59.6)	0.313
Deceased	81 (39.8%)	33 (43.4%)	0.587

Abbreviations: CR = complete response; PR = partial response; SD = stable disease; PD = progressive disease, ORR = objective response rate (CR + PR); ICI = immune checkpoint inhibitors; CI = confidence interval; TP = tumor progression; PTT = partial thromboplastin time; VTE = venous thromboembolic events; ATE = arterial thromboembolic events, including myocardial infarction, ischemic stroke, or acute arterial occlusion. ^1,2,3,4,5^ Statistics based on the total number of patients with known Breslow thickness (*n* = 207), ulceration status (*n* = 186), BRAF-status (*n* = 275), LDH serum levels (*n* = 238), and PTT at ICI initiation (*n* = 229). ^6^ Statistics based on the total number of patients with known BOR to initial ICI (*n* = 276); ^7,8^ The observation period for VTE/ATE started at the first day of initial ICI therapy and terminated either with the initiation of subsequent antineoplastic therapies or 3 months after the last cycle of initial ICI therapy, which is in accordance with clinical trials investigating ICI with a median time of reporting adverse events after the last cycle of 90 days [26,27]. ^9,10^ Statistics based on the total number of patients who received a re-induction of ICI therapy (*n* = 107) or BRAF/MEKi therapy (*n* = 64). The *p*-value is indicated in bold numbers when statistically significant. Bold is used in order to indicate titles or subtitles.

**Table 2 cancers-13-05103-t002:** Response to immune checkpoint inhibitor therapy based on the status of concomitant anticoagulation.

Outcome	Not on Anticoagulation	On Anticoagulation	*p*-Value
Best overall response—no. (%)			0.576
Complete response (CR)	30 (14.7%)	9 (12.3%)	
Partial response (PR)	48 (23.6%)	23 (31.5%)	
Stable disease (SD)	45 (22.2%)	13 (17.8%)	
Progressive disease (PD)	80 (39.4%)	28 (38.4%)	
Objective response rate ^1^			0.270
No. (%)	78/203 (38.4%)	32/73 (43.8%)	
95% CI ^1^	32.0–45.9%	32.2–55.9%	
Disease control rate ^2^			0.946
No. (%)	123/203 (60.5%)	45/73 (61.6%)	
95% CI ^3^	54.1–68.0%	49.5–72.8%	
Progress during initial ICI			0.647
No. (%)	159 (77.9%)	61 (80.2%)	
95% CI ^3^	71.6–83.4%	69.5–88.5%	

Abbreviations: Objective response rate was defined as the percentage of patients who obtained CR or PR; disease control rate was defined as the percentage of patients who obtained CR, PR, or SD. ^1,2^ ORR and DCR based on the total number of patients with known BOR to initial ICI (*n* = 203 for patients not on AC and *n* = 73 for patients receiving AC). ^3^ The 95% confidence intervals were calculated using the Clopper–Pearson method.

**Table 3 cancers-13-05103-t003:** Response to immune checkpoint inhibitor therapy stratified by the class of the anticoagulants used.

Outcome	Not on AC	Heparins	VKA	FXa DOAC	Chi-Square Test
Best overall response—no. (%)					**0.037**
Complete response (CR)	30 (14.7%)	1 (3.7%)	1 (5.0%)	7 (26.9%)	
Partial response (PR)	48 (23.6%)	5 (18.5%)	7 (35.0%)	11 (42.3%)	
Stable disease (SD)	45 (22.2%)	5 (18.5%)	4 (20.0%)	4 (15.4%)	
Progressive disease (PD)	80 (39.4%)	16 (55.2%)	8 (40.0%)	4 (15.4%)	
Could not be evaluated	3	2	0	1	
Objective response rate ^1^					**0.005**
No. (%)	78 (38.2%)	6 (22.2%)	8 (40.0%)	18 (69.2%)	
95% CI ^3^	32.0–45.6%	8.6–42.3%	18.1–63.9%	48.2–85.7%	
Disease control rate ^2^					**0.011**
No. (%)	123 (60.3%)	11 (40.7%)	12 (60%)	22 (84.6%)	
95% CI ^3^	54.1–68.0%	22.4–61.2%	36.1–80.9%	65.1–95.6%	
Progress during initial ICI					**0.001**
No. (%)	159 (77.9%)	28 (96.6%)	18 (90.0%)	15 (55.6%)	
95% CI ^3^	71.6–83.4%	82.2–99.9%	68.3–98.8%	35.3–74.5%	

Abbreviations: Objective response rate was defined as the percentage of patients who obtained CR or PR; disease control rate was defined as the percentage of patients who obtained CR, PR, or SD. VKA = vitamin K antagonists; FXa DOAC = Factor Xa dual oral anticoagulants. ^1,2^ ORR and DCR based on the total number of patients with known BOR to initial ICI (*n* = 203 for patients not on AC, *n* = 27 for patients receiving heparins, *n* = 20 for patients on VKA, and *n* = 26 for patients on FXa DOAC). ^3^ The 95% confidence intervals were calculated using the Clopper–Pearson method. The *p*-value is indicated in bold numbers when statistically significant.

**Table 4 cancers-13-05103-t004:** Summary of bleeding complications during initial ICI treatment stratified by the concomitant use of anticoagulants. Fisher’s exact test revealed that the risk of bleeding events during concomitant AC was not significantly higher compared to patients not receiving AC. Notably, the event of bleeding complications did not differ significantly between the particular agents of AC (heparin: *n* = 3; VKA: *n* = 2; FXa-i: *n* = 2, *p* = 0.923).

Secondary Outcome	Not on Anticoagulation	On Anticoagulation	*p*-Value
Bleeding complications			0.610
No. (%)	14 (6.9%)	7 (9.2%)	
95% CI	3.8–11.2%	3.8–18.1%	

Abbreviations: Bleeding outcomes included major and clinically relevant minor bleeding.

## Data Availability

All relevant data are within the manuscript and its supporting tables and figures.

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
