# Peer review of "Anticoagulation with Factor Xa Inhibitors Is Associated with Improved Overall Response and Progression-Free Survival in Patients with Metastatic Malignant Melanoma Receiving Immune Checkpoint Inhibitors—A Retrospective, Real-World Cohort Study"

_cancers, 2021, doi:10.3390/cancers13205103_

Round 1

Reviewer 1 Report

In this paper, the authors report on an observed statistical association between use of FXa inhibitors and ORR and PFS, but not OS, in a retrospective medical chart review of 280 patients treated with various types of immunotherapy for metastatic malignant melanoma over a 10-year period at one institution. No prognostic impact was found regarding the use of anticoagulants in general. Bleeding complications, ORR and PFS were assessed by chart review as were treatment factors and prognostic factors.

76 patients were retrospectively identified to be on anticoagulation, defined as treatment with anticoagulation for at least 1 month in the course of initial ICI therapy, whereof 27 patients were treated by FXa inhibitors. The reason for treatment with FXa inhibitors was for 13 patients (=48%) atrial fibrilation, while other classes of anticoagulants were perscribed primarily for tromboembolic events (Supplementary Table 5). Anticoagulant use was associated with older age (70 vs 63 yrs) and less use of combination ICI therapy. Treatment with FXa inhibitors in particular was only statistically significantly associated with older age (69 vs 63 yrs).

WHO performance status of patients was not reported. Year of treatment initiation was not included in the analysis, despite data suggesting an increased use of FXa inhibitors over time.  

The assessment of bleeding complications was a secondary end point. Only 21 bleeding complications were documented during initial ICI therapy, and no significant associations with use of anticoagulants were found. The authors suggest that patients with bleeding events might have been primarily treated elsewhere (not indentified by the single institutional chart review), and state that "interpretation of our data on bleeding complications requires caution".

The authors state that patients given AC and especially FXa inhibitors trend to represent a cohort with negative prognostic factors, but this was not supported by their data and for FXa-inhibitors this statement is not intuitive. 

Looking at Figure 3A it is striking that the difference in PFS-probability curves is due to a large difference in PFS-probability observed at time=0. The curve starts at almost 1.0 for the group treated by FXa-inhibitors, but starts at 0.5-0.75 for the other groups. Hence, 25%-50% of patients not treated by FXa inhibitors had a PFS-event at time= approx. 0. The authors do not address this crucial point, which may be explained by the definition of "beeing on" anticoagulants and/or the reasons for non-FXa inhibitor anticoagulant treatment beeing associated with a progression-event. 

The small N (=29) does not allow for sufficient elucidation of potential important confounders. One might for example suggest that FXa inhibitors are used primarily for atrial fibrilation and minor tromboses with little negative prognostic impact and that the likely increase in use of FXa inhibitors over time correlates with introduction of more efficient treatment regimens.

Author Response

Point-to-Point-answers to the reviewers´ comments:

Reviewer#1:

  1. WHO performance status of patients was not reported. Year of treatment initiation was not included in the analysis, despite data suggesting an increased use of FXa inhibitors over time.

Ad 1: We thank the reviewer for this important hint that adding the year of anticoagulant treatment initiation might allow for a more precise comparison of the patient subcohorts receiving different anticoagulants. We have therefore complemented our analysis with this new data (see section 3, P14; L364-368 and supplementary material Table 5): Indeed, the use of FXa-inhibitors has increased over time (median: July 2018), while heparin treatment showed a trend to be rather initiated in the early observation period (median: March 2017), albeit this association was below statistical significance (two-sided p=0,123 in Kruskal-Wallis-test). Patients receiving no anticoagulation at all (median: December 2017) and patients receiving Vitamin-K antagonists (median: February 2018) did also not significantly differ in terms of the date of AC treatment initiation when compared to FXa-inhibitors. Unfortunately, WHO performance status could not be assessed in our retrospective analysis, since ECOG performance status has not been systematically documented in our institution until late 2015. Therefore, we have refrained from including this statistical parameter in our analysis.

  1. The assessment of bleeding complications was a secondary end point. Only 21 bleeding complications were documented during initial ICI therapy, and no significant associations with use of anticoagulants were found. The authors suggest that patients with bleeding events might have been primarily treated elsewhere (not indentified by the single institutional chart review), and state that "interpretation of our data on bleeding complications requires caution".

Ad 2: We share the concern of the reviewer that our single institutional chart review might not sufficiently allow for a comprehensive analysis of the patients´ bleeding complications and that bleeding complications might therefore be underestimated. Due to the small number of bleeding complications, we suspected that patients with bleeding complications and acute symptoms might be primarily treated and documented in clinical units (or even other hospitals) other than the Department of Dermatology, which complicates the systematic documentation of bleeding events in our retrospective study. Although the assessment of bleeding complications and treatment-associated side effects is a common scheme in the Department of Dermatology, a comprehensive documentation of side-effects and bleeding complications also requires a strong awareness for and documentation of these complications among external physicians (section 4; P17, L504-513). Therefore, this issue certainly requires further investigation in prospective clinical trials which should systematically analyze the incidence of bleeding complications during ICI therapy and concomitant AC.

  1. The authors state that patients given AC and especially FXa inhibitors trend to represent a cohort with negative prognostic factors, but this was not supported by their data and for FXa-inhibitors this statement is not intuitive. 

Ad 3: We have taken into account the valuable criticism of the reviewer that our data might not sufficiently explain our conclusion and therefore provided additional information supporting our argument that patients receiving anticoagulation in general, and FXa-DOACs in particular, tend to represent a cohort with negative prognostic factors. Here, we reasoned that patients who have not received anticoagulants less often presented with melanoma brain metastasis (no AC: 32.3% vs AC in general: 34.6% vs FXa-DOAC: 38.4%) or elevated LDH-serum levels (347.1 vs 369.7 vs 382.0), and more often were eligible for combined checkpoint-inhibitor therapy (43.6 vs 26.3 vs 33.3%), which is considered the treatment of choice in non-BRAF mutant patients, since cICB is associated with better rates of objective response, a longer PFS, sustained disease control and a longer OS. Furthermore, patients receiving anticoagulation also tend to represent a cohort with negative prognostic factors with regard to clinical comorbidities which might worsen the overall survival of patients. In particular, patients on anticoagulation might more often present with significant cardiovascular preconditions and are at higher risk of cardiovascular and thromboembolic events, such as VTE (4.4 vs 36.8 vs 29.6%) or ATE (2.9 vs 7.9 vs 8.3%). Overall, we have thus reasoned that it seems conceivable, that patients receiving AC in general and FXa-DOACs in particular, represent a cohort with negative prognostic factors as compared to patients not receiving AC at all (P17; L489-503).

On the other hand, we agree with the reviewer that our data do not provide reasonable evidence for arguing that patients receiving FXa-DOACs might show worse prognostic factors as compared to patients receiving other anticoagulants, such as heparins or VKA. Although patients receiving FXa DOACs were at higher age at the initiation of ICI and presented with thicker melanomas (3.4mm vs 2.1mm vs 3.4mm) and higher LDH serum levels (382.0 vs 385.8 vs 322.7), these patients more often received ICI therapy in a first-line setting (85.2% vs 55.2% vs 80.0%, p=0.03), more often received combined ICI therapy (33.3% vs. 24.1% vs 20.0%, p = 0.61) and there was a trend that the use of ICI increased over time for patients on FXa-DOACs. As the reviewer has already pointed out, patients on FXa-DOACs might also show a better risk profile in terms of cardiovascular preconditions as compared to patients receiving heparins or VKA, since these patients did mainly receive anticoagulation following a major thromboembolic event such as LAE or DVT, whereas patients on FXa-DOACs received AC mainly for atrial fibrillation or minor DVT. Therefore, patients on FXa-DOACs might show better prognostic factors as compared to patients receiving heparins or VKA, yet worse prognostic factors as compared to patients not receiving AC at all (P17; L496-503).

  1. Looking at Figure 3A it is striking that the difference in PFS-probability curves is due to a large difference in PFS-probability observed at time=0. The curve starts at almost 1.0 for the group treated by FXa-inhibitors, but starts at 0.5-0.75 for the other groups. Hence, 25%-50% of patients not treated by FXa inhibitors had a PFS-event at time= approx. 0. The authors do not address this crucial point, which may be explained by the definition of "beeing on" anticoagulants and/or the reasons for non-FXa inhibitor anticoagulant treatment beeing associated with a progression-event. 

Ad 4: We have taken into account the valuable criticism and addressed this important concern raised by the reviewer. Indeed, we have observed significantly more PFS events at the approx. time 0-1 months in the patient subcohorts who have received concomitant treatment with heparins, VKA, or patients not being on AC as compared to patients receiving FXa-DOACs, which explains the differences in the PFS probability curves in the Kaplan Meier plots of Figure 3A. With regard to the patient subcohort receiving heparin treatment, the significantly increased occurrence of PFS events at this early time point can be explained by the chronological proximity or the co-occurrence of severe, and fatal, thromboembolic events at the time of initiation of ICI therapy (such as the occurrence of pulmonary embolism). On the other hand, patients on FXa-DOACs, largely received anticoagulants following minor thrombotic events (i.e. DVT, or atrial fibrillation), which are unlikely to be associated with a severe or fatal course.

We have therefore comprehensively discussed this crucial point: In particular, we share the concern of the reviewer that the occurrence of fatal thromboembolic events might be considered an important confounder in the PFS analysis, as these might impact the PFS curves irrespective of the biological response to immune-checkpoint blockade (P17; L479-482).    

  1. The small N (=29) does not allow for sufficient elucidation of potential important confounders. One might for example suggest that FXa inhibitors are used primarily for atrial fibrilation and minor tromboses with little negative prognostic impact and that the likely increase in use of FXa inhibitors over time correlates with introduction of more efficient treatment regimens.

Ad 5: We agree with the reviewer that FXa-inhibitors were indeed primarily used for atrial fibrillation and DVT. As compared to patients receiving heparins following acute and severe TEE, such as pulmonary embolism or major DVT, patients on FXa DOACs might therefore certainly show a better risk profile in terms of cardiovascular diseases. Since major TEE, such as pulmonary embolism, may yet be considered life-threatening events, these events certainly represent important confounders, which should be considered in our analysis. We have therefore complemented our discussion with these important data (P17; L494-501), as previously explained. Furthermore, we share the concern of the reviewer that the small number of patients in each subcohort receiving concomitant treatment with different anticoagulants complicates to sufficiently elucidate the statistical effect of these important confounders (in our analysis, we could not find statistically significant differences between the 3 patient subcohorts, p=0.34). We have therefore comprehensively discussed the role of the different areas of application for the different anticoagulants and the impact of these potential confounders on survival analysis in order to account for this concern. We have further included our observation, that the use of FXa DOACs did non-significantly increase over time, which might yet correlate with the introduction of more efficient treatment regimens and might therefore be considered another potential confounder (P17; L501-503).   

Reviewer 2 Report

The article ’Anticoagulation with Factor Xa inhibitors is associated with improved overall response and progression-free survival in patients with metastatic malignant melanoma receiving immune checkpoint inhibitors – a retrospective, real-world cohort study’ by Haist & Stege et al presents retrospectively collected data evaluating the prognostic effects of AC treatment co-administered with ICI in a large cohort of melanoma patients.

The authors found no survival association with AC treatment. However, they found that a particular class of AC – Fxa inhibitor did show significant, prognostic effects when compared to no/other AC inhibitors. Several clinical variables (BRAF, LDH etc) and potential confounders were analyzed.

Overall comments:

Overall, the topic is clinically relevant, the manuscript is well written, and the bio-statistical analyses is appropriate. Minor typos need to be corrected as indicated below. Based on the graphical abstract I expected more biological characterization.

Specific comments:

  1. The graphical abstract is not described. If the authors find this important, it should be described and addressed in the main text. The graphical abstract implies deeper biological characterization, which is not done here. The authors could expand their discussion specifically focusing on what the graphical abstract depicts or add in silico data.
  2. Starting from line 282 to 288 it is unclear which data/table/figure the authors are referring to.
  3. On lines 312 and 334 Figure 5A and Figure 5B are cited, however no such figure exists. The text describes Figure 3.
  4. In the main text and in the label of ‘Table 5’ should be changed to Table 4.
  5. Minor English spell check and correction is required.

Author Response

Point-to-Point-answers to the reviewers´ comments:

Reviewer #2:

  1. The graphical abstract is not described. If the authors find this important, it should be described and addressed in the main text. The graphical abstract implies deeper biological characterization, which is not done here. The authors could expand their discussion specifically focusing on what the graphical abstract depicts or add in silico data.

Ad 1: We thank the reviewer for this valuable hint and have expanded the description of the biological background underlying the effects observed in our retrospective clinical study. In particular, we have explained in more detail the postulated mechanisms which allow for the restoration of anti-tumor immunity within the TME for patients receiving FXa DOACs. These mechanisms include the recruitment of cytotoxic lymphocytes and antigen-presenting dendritic cells, as well as preventing macrophage polarization towards an immunosuppressive M2 phenotype and preventing the infiltration of immunosuppressive regulatory T cells. Moreover, we have provided a more focused description of how macrophage-derived FXa contributes to immune evasion via PAR2. In contrast to other anticoagulants such as heparins (which are restricted to the intravascular space due to limited plasma leakage) or VKA, FXa inhibitors may specifically target these important signaling pathways (P16,17; L463-474).

Notably, however, our study primarily serves as a clinical investigation, which provides first clinical evidence for the preclinical observations reported by Graf et al and Metelli et al. Due to the retrospective nature of our analysis, we were not able to generate e.g. blood samples or tumor tissue for translational analysis of PBMCs, cytokine analysis, RNA-sequencing analysis etc., which would have allowed to address the biological role of FXa-DOACs on the TME in translational analyses. Unfortunately, these highly interesting questions cannot be answered from our data, since many patients from this retrospective study have already deceased, but will be the subject of further studies currently prepared in our laboratory.

  1. Starting from line 282 to 288 it is unclear which data/table/figure the authors are referring to.

Ad 2: We thank the reviewer for this valuable hint and have added the corresponding references to the supplementary tables and figures into this section in order to improve the readability and allow for a precise assignment of the data (P8; L275-282).

  1. On lines 312 and 334 Figure 5A and Figure 5B are cited, however no such figure exists. The text describes Figure 3.

Ad 3: We thank the reviewer for this important hint and have corrected the corresponding references. Indeed, we have here referred to Figure 3, which illustrates the survival curves stratified by the specific agent of anticoagulation received by the patient subcohorts (P10; L314 and P12; L336).

  1. In the main text and in the label of ‘Table 5’ should be changed to Table 4.

Ad 4: We have corrected this tipping mistake and have changed the annotation to Table 4 (P15; L401)

  1. Minor English spell check and correction is required.

Ad 5: We thank the reviewer for this valuable criticism and have thoroughly conducted an additional spell and correction check and hope to meet the journal's required standards.

Reviewer 3 Report

This is an excellent and important paper. I only suggest that the sentence beginning on line 512 and ending on line 515 be deleted.

It could be replaced by the suggestion that concomitant FXa DOAC treatment be the subject  of additional clinical studies by others. 

Author Response

Point-to-Point-answers to the reviewers´ comments:

Reviewer #3:

  1. This is an excellent and important paper. I only suggest that the sentence beginning on line 512 and ending on line 515 be deleted. It could be replaced by the suggestion that concomitant FXa DOAC treatment be the subject  of additional clinical studies by others. 

Ad 1: We thank the reviewer for the kind assessment of our manuscript and agree with the reviewer that the assessment of the clinical efficacy of concomitant FXa DOAC treatment should rather be subject of additional clinical studies, which might contribute to the evidence presented in this study, before being introduced into broad clinical practice. We have therefore changed the sentence accordingly (P18; L534-543). 

Reviewer 4 Report

The manuscript can be accepted with minor points to address.

  1. Please increase font on axes for Kaplan Myers plots, add p-values to each plot
  2. Expand Discussion with adding more other data supporting this study

Author Response

Point-to-Point-answers to the reviewers´ comments:

Reviewer #4:

  1. Please increase font on axes for Kaplan Myers plots, add p-values to each plot

Ad 1: We thank the reviewer for the kind assessment of our research article and agree that the font on the axes of the Kaplan Meier plots needs to be increased in order to allow for a better illustration of our data. We have therefore increased the font on the axes and added p-values to each plot (see section 3: P10; L291 and P14; L344; and see supplementary Material: supplementary Figures 1 and 2).  

  1. Expand Discussion with adding more other data supporting this study

Ad 2: We have taken into account the valuable criticism of the reviewer and complemented our discussion with additional data regarding the time of AC treatment initiation (supplementary Table 5), the clinical characteristics of the different subcohorts receiving concomitant anticoagulation, and statistical analysis in order to provide comprehensive insights into our supporting data and adequately explain our conclusions (see section 4: P17, L489-503).

Round 2

Reviewer 1 Report

Your main hypothesis is highly interesting and, although confounding cannot be excluded in this retrospective chart review study, prospective studies are warranted